# The Role of C2 Domains in Two Different Phosphatases: PTEN and SHIP2

**DOI:** 10.3390/membranes13040408

**Published:** 2023-04-04

**Authors:** Laura H. John, Fiona B. Naughton, Mark S. P. Sansom, Andreas Haahr Larsen

**Affiliations:** 1Department of Biochemistry, University of Oxford, Oxford OX1 3QU, UK; 2Department of Neuroscience, University of Copenhagen, 2200 Copenhagen, Denmark

**Keywords:** PTEN, SHIP2, phosphoinositol, PIP, MD, C2 domain, phosphatase

## Abstract

Phosphatase and tensin homologue (PTEN) and SH2-containing inositol 5′-phosphatase 2 (SHIP2) are structurally and functionally similar. They both consist of a phosphatase (Ptase) domain and an adjacent C2 domain, and both proteins dephosphorylate phosphoinositol-tri(3,4,5)phosphate, PI(3,4,5)P_3_; PTEN at the 3-phophate and SHIP2 at the 5-phosphate. Therefore, they play pivotal roles in the PI3K/Akt pathway. Here, we investigate the role of the C2 domain in membrane interactions of PTEN and SHIP2, using molecular dynamics simulations and free energy calculations. It is generally accepted that for PTEN, the C2 domain interacts strongly with anionic lipids and therefore significantly contributes to membrane recruitment. In contrast, for the C2 domain in SHIP2, we previously found much weaker binding affinity for anionic membranes. Our simulations confirm the membrane anchor role of the C2 domain in PTEN, as well as its necessity for the Ptase domain in gaining its productive membrane-binding conformation. In contrast, we identified that the C2 domain in SHIP2 undertakes neither of these roles, which are generally proposed for C2 domains. Our data support a model in which the main role of the C2 domain in SHIP2 is to introduce allosteric interdomain changes that enhance catalytic activity of the Ptase domain.

## 1. Introduction

C2 domains are structurally conserved modules of about 130 residues, which fold as beta-sheet sandwiches [1]. C2 domains are often considered as membrane anchors but may play alternative or additional roles in membrane-located signaling proteins. The phosphatase and tensin homologue (PTEN) and SH-containing inostitol 5-phospatase 2 (SHIP2) are both phosphatases. They function by dephosphorylating phosphoinositol-tri(3,4,5)phosphate, PI(3,4,5)P_3_ to form PIP_2_ [2] (Figure 1). The product of PTEN dephosphorylation is phosphoinositol-di(4,5)phospate (PI(4,5)P_2_), whereas SHIP2 forms PI(3,4)P_2_. This process is part of several signaling cascades [3]. The presence of PIP_3_ promotes proliferation of the cell, so by removing PIP_3_ from the membrane, PTEN acts as a tumor suppressor, and mutations of the PTEN gene are associated with cancer [4].

Both PTEN and SHIP2 consist of an N-terminal phosphatase (Ptase) domain, followed by a C2 domain. The Ptase domain contains the active sites, whereas the C2 is an auxiliary domain and is widely believed to facilitate membrane binding in SHIP2 and PTEN, as well as in several other protein classes [5]. Therefore, we were surprised by our previous coarse-grained molecular dynamics (MD) simulations, which showed that the C2 domain of SHIP2 (C2_SHIP2_) bound only weakly to a lipid bilayer mimicking mammalian inner membrane [6]. In all-atom MD, C2 _SHIP2_ even dissociated from the membrane, verifying the weak lipid binding. Those preliminary results challenge the conception of C2 as a membrane anchor, at least in the case of SHIP2. In this study, we therefore investigated the role of C2_SHIP2_ and compared it with the corresponding domain of the related protein PTEN. We used MD simulations to study the protein complexes containing both Ptase and C2 domains (denoted here as full-length PTEN or SHIP2), as well as the isolated Ptase domains. The results can be directly compared with the previous simulations of isolated C2 domains of the two proteins [6], as we used the same simulation setup, i.e., lipid composition, simulation duration, number of repeats, etc. (see Section 2).

**Figure 1 membranes-13-00408-f001:**
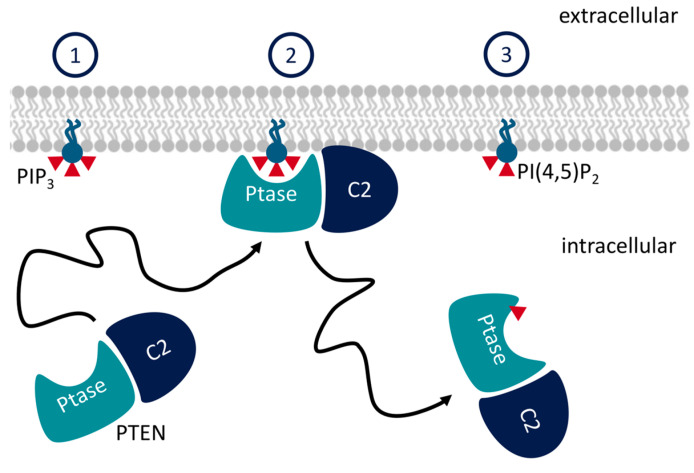
The function of PTEN: PTEN binds the inner leaflet of plasma membrane and dephosphorylates PI(3,4,5)P_3_ to form PI(4,5)P_2_. SHIP2 works similarly, but the product is instead PI(3,4)P_2_. Adapted from Ref. [7].

## 2. Methods

### 2.1. Preparing Initial Structures: Addition of Missing Residues

The full-length PTEN consisted of residues 1–403, whereof the structure of residues 14–281 and residues 312–351 was taken from the X-ray crystal structure (PDB 1D5R) [8]. The missing N-terminal residues 1–13 were modeled as a helix, using Modeler [9]. Missing residues 282–311 and C-terminal residues 352–403 were modelled as loops, also using Modeler [9].

We previously simulated lipid binding of a Ptase-C2 core of PTEN [6], but that was without the N-terminal helix. Ptase_PTEN_ consisted of residues 1–186 and C2_PTEN_ of residues 191–352 of the full-length construct.

The full-length SHIP2 was simulated previously [6] and consisted of residues 418–878. The structure of residues 420–731 and residues 745–874 was taken from chain B of the X-ray crystal structure (PDB 5OKM) [10]. The missing residues 418–419, 732–744, and 875–878 were modeled as loops, using Modeler [9]. Ptase_SHIP2_ consisted of residues 418–729 and C2_SHIP2_ of residues 747–862 of the full-length construct.

### 2.2. Coarse-Grained MD Simulations

The initial structures were rotated through a random angle, then centered before they were transformed into a coarse-grained model following the Martini 2.2 coarse-graining scheme [11], using the *martinize* script [12]. An elastic network was applied to the protein with default settings, and a secondary structure was given as input using the DSSP algorithm [13]. The protein was inserted 4.4 nm above a lipid bilayer using the *insane* script [14]. The membrane consisted of 80% POPC, 15% POPS, and 5% POPIP_2_ (Martini lipid POP2 [15]) (see lipid topologies at www.cgmartini.nl). The lipid headgroup is based on PIP(3,4)P_2_ [15]. The insane script also solvated the protein and membrane in 90% Martini water, with 10% antifreeze particles [11] and neutralized the system with Na^+^ or Cl^−^ ions. For Ptase_SHIP2_ simulations, the box was 10 × 10 × 20 nm^3^; for Ptase_PTEN_ simulations, the box was 7 × 7 × 18 nm^3^; and for full-length simulations, the box was 12 × 12 × 24 nm^3^. The simulations were performed using the Martini 2.2 forcefield [12]. The use of Martini 2.2 made it possible to compare the results directly with previous simulations [6]. The configuration was minimized with a steepest descend algorithm. Van der Waals and Coulomb cut-offs were both 1.1 nm, so four times less than the initial distance between protein and membrane. The system was equilibrated for 10 ns with a 20 fs timestep in the NPT ensemble, with a semi-isotropic Berendsen barostat [16] set at 1 bar with a time constant of 1 ps and compressibility of 0.3 kbar^−1^; the v-scale thermostat [17] was set at 323 K, with a time constant of 1 ps. The simulations were constrained using the LINCS algorithm [18]. After equilibration, simulations were run for 2 μs with a 35 fs time step and the same settings as the equilibration. All coarse-grained simulations were repeated 25 times.

### 2.3. All-Atom MD Simulations

The coarse-grained binding poses were converted to all-atom models using CG2AT [19] (version 0.2). The all-atom simulations were run with the CHARMM36m force field [20] with TIP3P water. Cut-off distances for van der Waals and Coulomb forces were 1.2 nm. Long-range electrostatics were handled by the Particle Mesh Ewald algorithm [21]. The system was first minimized and then equilibrated in the NVT ensemble for 100 ps with a 2 fs time step. A v-rescale thermostat [17] was applied to keep the temperature at 300 K, with a time step of 0.1 ps. The system was subsequently equilibrated for 100 ps with a 2-fs time step in the NPT ensemble, using a Parrinello–Rahman semi-isotropic barostat [22] to keep pressure at 1 bar, with a time constant of 5 ps and compressibility of 0.045 kbar^−1^. In both equilibration simulations, the protein atoms were position restrained by a force constant of 1000 kJ/mol in all directions. Finally, a 600 ns production run was generated with a 2 fs time step and without position restraint on the protein.

### 2.4. Analysis of Coarse-Grained Simulations

The distance between the protein center of mass and the membrane center of mass (after centering of the protein in the box) was calculated with the GROMACS algorithm *gmx distance*. The *zz* component of the rotational matrix (*R_zz_*) was calculated with *gmx rotmat*. Binding poses were selected from the frequency densities in the *R_zz_* vs distance 2D landscapes.

### 2.5. Analysis of All-Atom MD Simulations

The center-of-mass distance was calculated with the *gmx distance* tool and minimum distances by *gmx mindist*. Root mean square deviations (RMSD) were used to monitor the difference between the binding pose of isolated Ptase and Ptase in the full-length Ptase–C2 complex bound to the same lipid membrane. The RMSD were calculated after alignment in the *xy*-plane (i.e., the approximate bilayer plane) and around the *z*-axis using an in-house script. RMSD was also used to monitor the structural change after converting the coarse-grained binding pose to all-atom structure and simulating it in the CHARMM-36 m all-atom forcefield [20] with TIP3P water. In this case, the frames were compared with the initial frame in the atomistic simulation and RMSD were calculated with GROMACS using *gmx rms*. Finally, RMSD were used to compare the SHIP2 full-length and the Ptase_SHIP2_ simulation with a previously reported crystal structure, respectively [10]. For this, the RMSD were calculated per residue as time averages over the 600 ns simulations, with the initial frame as reference structure using *gmx rmsf*. For the same simulations, root mean square fluctuations (RMSFs) were used to monitor fluctuations of regions in the Ptase domain of SHIP2 during the 600 ns simulations. These were likewise calculated with *gmx rmsf*.

### 2.6. Free Energy Calculations

Binding poses from the coarse-grained simulations were taken as initial structures. These were then pulled away from the membrane along the *z*-coordinate, using the GROMACS pull code. A pull rate of 0.2 nm/ns with a harmonic (umbrella-type) constraint with a force constant of 1000 kJ/mol/nm^2^ was used. The binding pose was likewise pushed 0.24 nm closer to the membrane center, with the same settings, except that the rate was inverted to be −0.2 nm/ns. Frames with a distance of 0.05 nm were taken from the push and pull trajectories, ranging from the frame where the protein was pushed 0.24 nm closer to the membrane center of mass than the binding pose to a frame where the protein was pulled to a distance of 7 nm (for Ptase simulations) or 8.5 nm (for full-length simulations) from the membrane center of mass. Position restraints were applied on all lipids in the pull simulations (not in the push simulations or umbrella simulations) to prevent lipids being pulled out of the membrane together with the protein. Each frame (window) was sampled for 1 μs with a time step of 35 fs, without position restraint on the lipids. The potential of mean force (PMF) was calculated from the forces using the WHAM algorithm [23], as implemented in GROMACS by the command *gmx wham* [24], omitting the first 100 ps from the analysis. Bootstrapping using 200 repeats was used to estimate the standard deviation [24].

We wished to obtain the free energy contributions for C2 and Ptase in the full-length binding mode, respectively. To this end, Ptase and C2 from full-length PTEN were extracted, and each domain was then placed back by the membrane and resolvated. The same was carried out for Ptase and C2 from SHIP2. The GROMACS function *gmx trjconv* was used for extraction. Resolvation was performed with insane using the same box size as the box from which the protein was extracted. Topology files were changed manually to accommodate the changes. The system was minimized and equilibrated in the NPT ensemble for 10 ns, with a time step of 30 fs and with the protein restrained by a 1000 kJ/mol/nm^2^ force constant. Umbrella sampling was then performed as described above.

### 2.7. Free Energy Surfaces, Visualization, and Plotting

Free energy surfaces were generated with PyMOL (vacuum electrostatics). Structures were visualized using PyMOL (The PyMOL Molecular Graphics System, Version 1.2r3pre, Schrödinger, LLC, New York, NY, USA), and plots were generated in Python3, using Numpy [25] and Matplotlib [26].

### 2.8. Reproducibility and Data Availability

Scripts generated for running the simulations are made available at GitHub: https://github.com/andreashlarsen/John2023-PTEN_SHIP2 (accessed on 23 March 2023).

## 3. Results

### 3.1. N-Terminal Helix in PTEN Participates in Lipid Binding

Initially, we focused on full-length PTEN and SHIP2. We have simulated these constructs previously [6], but we made new simulations for full-length PTEN to include an N-terminal helix in Ptase_PTEN_, which was not modeled in the PTEN structure in ref [6]. Twenty-five coarse-grained simulations were performed on full-length PTEN with the N-terminal helix. The simulations were performed with an anionic membrane that mimics the inner leaflet of mammalian plasma membrane, as PTEN and SHIP2 bind to this membrane in vivo [6]. To map out the “binding landscape”, we plotted the center of mass–distance from membrane to C2 domain against *R_zz_*. *R_zz_* is the zz-component of the rotation matrix with respect to a reference structure, in this case, the last frame of the first repeat. Thus, *R_zz_* represents a difference in binding orientation. Two binding modes were observed (Figure 2A), but only one was productive, i.e., with the catalytic active site of the Ptase domain in contact with the membrane [6]. The N-terminal helix contributed to the lipid binding in the productive binding mode by insertion into the lipid bilayer (Figure 2B). The productive binding mode was, however, very similar to the productive mode for simulations of PTEN without the N-terminal helix (Figure 2B) (from Ref. [6]), except for a tilt that allowed deeper helix insertion.

### 3.2. Membrane Binding of Isolated Ptase from PTEN and SHIP2

We performed 25 repeats of 2 μs coarse-grained simulations of Ptase_PTEN_ to investigate the different lipid binding modes of Ptase in the absence of C2. Using these simulations, the binding landscape could be mapped out, as carried out for full-length PTEN (Figure 2A). However, this time, the Ptase conformation from the productive binding mode of the full-length PTEN was used as reference structure. Therefore, *R_zz_* = 1 means that the simulated Ptase domain has the same orientation as the Ptase domain from the full-length binding mode, whereas *R_zz_* = −1 corresponds to a 180° rotation. Three binding modes were observed (Figure 3A,C). None of the observed modes were identical to the Ptase of the full-length PTEN simulation (Figure 3D), with a minimum RMSD of 0.9 nm for mode 3.

We also performed 25 repeats of 2 μs coarse-grained simulations of Ptase_SHIP2_ and compared the binding conformations with the Ptase binding conformation of the full-length SHIP2 simulation [6]. In this case, we only observed two binding modes (Figure 3B,E). Mode 2 resembled that of Ptase from the full-length SHIP2 closely (Figure 3F), with an RMSD of 0.2 nm.

### 3.3. Free Energy Calculations

From the CG simulations of full-length PTEN, we determined a productive binding mode, where the active sites of the Ptase domain were in contact with the membrane (Figure 2). Several frames from the CG simulations fulfilled that criterion (Mode 1 in Figure 2A), and the frame with the most frequently occurring values of *R_zz_*, and protein–lipid distance was selected (yellow area in Figure 2A). The same approach was used to select the productive binding mode for full-length SHIP2 [6].

The productive binding modes of PTEN and SHIP2 (Figure 3D,F) were used for free energy calculations. First, we performed umbrella sampling and potential of mean force (PMF) calculations for the unbinding of full-length PTEN and SHIP2 from the lipid membrane. The binding energy (ΔPMF) was calculated as the difference between the PMF minimum and the PMF in bulk (Figure 4). The binding energy of PTEN was estimated to be −162 ± 1 kJ/mol, and the binding energy of SHIP2 was estimated to be substantially smaller, namely −96 ± 2 kJ/mol.

We also analyzed the contribution from C2 and Ptase, respectively, to the binding energy of the full-length complexes. To this mean, we first removed C2 from the full-length PTEN productive binding mode, so only Ptase was left in its binding mode. We then calculated the free energy of unbinding for the Ptase domain. The same was conducted for all four domains, i.e., for C2 and Ptase from PTEN and SHIP2 (Figure 4B). For PTEN, the Ptase domain and the C2 domain contributed equally to the total binding energy. For SHIP2, on the other hand, the contribution from Ptase to the total binding energy was almost three-fold higher than the contribution from the C2 domain.

The energy contribution from Ptase_PTEN_ was 103 kJ/mol, and the contribution from C2_PTEN_ was 109 kJ/mol (only magnitudes are given for simplicity; therefore, the sign is omitted). This adds up to 212 kJ/mol, i.e., 30% more than the estimated free energy of the binding of full-length PTEN, which was 163 kJ/mol. Correspondingly, for SHIP2, the sum of the free energies from the components was 105 kJ/mol, which is 9% more than the estimated free energy of the binding of full-length SHIP2, which was 96 kJ/mol. The higher degree of freedom of movement for the single domains (which are not constrained by neighboring domain) may lead to this difference through higher entropy and more optimal alignment with the membrane.

The difference in binding energy between the C2 domains of SHIP2 and PTEN is reflected in the distribution of charged residues on their surface. C2_PTEN_ has a positive groove (dominated by Arg and Lys) facing the negatively charged membrane (Figure 4C), thus forming an attractive binding site for PIP_2_ lipids. C2_SHIP2_, on the other hand, has more negatively charged residues (Glu and Asp) facing the membrane, which prevent tight binding to the anionic membrane (Figure 4D).

The C2 domains of full-length PTEN and full-length SHIP2 have the same overall binding orientation with respect to the lipid membrane, with loop12 (the loop between helix 1 and helix 2), loop56, and loop78 facing the membrane and the N- and C-termini pointing away from the membrane. Loop56 of C2_PTASE_ contains five cationic residues (Lys) and only one anionic one (Asp) (H**K**QN**K**ML**KKDK**M), and loop78 contains four cationic residues (Lys and Arg) against two anionic residues (Asp) (**K**N**D**L**DK**AN**R**YFSPNF**K**). Although loop12 contains no cationic residues and one anionic residue (Glu) (F**E**TIPMFSGGTCN), the overall charge of the three loops of C2_PTEN_ is highly positive (+5e) and provides an attractive binding interface for anionic lipids, such as PIP_2_. In C2_SHIP2_, loop12 is cationic (**K**TAS**R**T**K**), but loop 56 (**K**SM**D**GY**E**SY) and loop78 (H**R**G**EE**) are slightly anionic, and the total charge of the three loops C2_SHIP2_ is only +1e.

The same overall binding orientation was observed for isolated C2 domains of PTEN and SHIP2 (i.e., without the Ptase domains), and this binding mode is typical for calcium-independent Type II C2 domains [6].

### 3.4. Atomistic Simulations of Full-Length Structures

We converted the full-length binding conformations of PTEN and SHIP2 to atomistic resolution and performed a 600 ns all-atom simulation. That is, the initial positions were determined by the outcome of the CG simulations (as described in Section 3.3), which in turn started from randomized initial orientations. For PTEN, we observed an RMSD from the first to last frame of about 0.8 nm (Figure 5A). From visual inspection, this comes partly from loops being less compact (Figure 5D), but we also observed that the C2 and Ptase domains of PTEN shifted away from each other (Figure 5C). Moreover, the center of mass of PTEN moved away from the membrane by about 0.5 nm (Figure 5B), indicating less membrane insertion. However, the protein stayed bound throughout the simulations, and the number of contacts between both domains and PIP_2_s in the membrane stayed constant (Figure 5F). A contact is defined here as a minimum distance of less than 0.6 nm [27].

For SHIP2, the RMSD between the first and last frame was minor, only 0.3 nm (Figure 5A). Moreover, there was only a minor shift away from the membrane, between 0.1 and 0.4 nm during the simulation (Figure 5B). Moreover, PIP_2_s stayed bound to the complex throughout the simulation, but notably, no PIP_2_ was bound to the C2 domain, except in the Ptase–C2 junction (Figure 5E,F), which could explain why this domain had the smallest contribution to the binding energy (Figure 4B).

### 3.5. Dynamics of Full-Length SHIP2 vs Ptase_SHIP2_

The Ptase_SHIP2_ constructive binding mode (Figure 3, mode 2) was also converted to all-atom resolution and simulated for 600 ns, to compare the dynamics of Ptase_SHIP2_ alone and in the presence of C2_SHIP2_ (Figure 6). The dynamics were quantified in terms of the

RMSD and the RMSF, averaged over the 600 ns simulation. RMSD were calculated with respect to the initial frame, which corresponds to the final coarse-grained binding conformation, converted into an all-atom resolution. Having had an elastic network applied during the coarse-grained simulations, the overall conformation of this reference structure is close to the initial crystal structure (PDB: 5OKM). Previous MD simulations showed that C2_SHIP2_ affects the conformation of the loop L4 (residues 674–684) and helices α5-7 (residues 615–643) of Ptase_SHIP2_ [10]. These simulations were performed without a lipid membrane, so we tested whether the results could be reproduced in the presence of a lipid membrane. Our simulations confirm this conformational change with SHIP2 bound to a membrane, as the RMSD of those regions vary in the presence or absence of C2_SHIP2_ (Figure 6A). We also confirm that loop L3 of Ptase_SHIP2_ (residues 587–594), which is in the interface between the domains, is less flexible in the presence of C2_SHIP2_, as shown from a decrease in RMSF (Figure 6B). However, C2_SHIP2_ does not change fluctuations of loop L4 and helices α5-7 significantly (Figure 6B), although this was reported in the simulations without a membrane [10]. Moreover, loop L2 of Ptase_SHIP2_ (residues 531–539) showed a strong reduction in fluctuations in the presence of C2_SHIP2_ (Figure 6B) in our simulations, as opposed to the simulations without a membrane, where the dynamics of L2 increased in the presence of C2 [10]. This is likely due to the fact that L2 is facing the membrane.

## 4. Discussion

In order to probe the different roles of C2 in the phosphatases PTEN and SHIP2, we measured their contributions to the free energy of lipid binding (Figure 4). In PTEN, the C2 and Ptase domains contribute equally to the lipid-binding energy, whereas in SHIP2, the binding energy is dominated by the Ptase domain. Moreover, PTEN exhibits a substantially stronger overall membrane binding than SHIP2.

### 4.1. Negative Net Charge of the Membrane Is Critical for C2 Domain Binding

We have previously simulated membrane binding of C2_PTEN_ and C2_SHIP2_ under the same conditions as in the present study, but with a smaller membrane (7 nm × 7 nm versus 12 nm × 12 nm) [6]. With the smaller membrane, the binding energy of C2_PTEN_ was estimated to 78 ± 4 kJ/mol (compared with 109 ± 1 kJ/mol with the larger membrane), and the binding energy of C2_SHIP2_ was estimated to 23 ± 1 kJ/mol (compared with 30 ± 1 kJ/mol with the larger membrane), i.e., the energies were estimated to be 30–40 % larger in the 12 nm × 12 nm membrane compared with the previously used 7 nm × 7 nm membrane. This may be due to the increased number of PIP_2_ lipids in the larger membrane (in absolute numbers, as molar concentration was 5% in both cases). There were 11 PIP_2_ lipids in each leaflet of the 12 nm × 12 nm membrane in the present study versus 4 in the 7 nm × 7 nm membrane [6]. This supports the known biological importance of polyphosphoinositides [28]. When PIP_2_ was not present (i.e., PC/PS in anionic membrane), the binding energies were estimated to be two–three-fold smaller [6]. Negative net charge of the membrane is pivotal, as C2 domains generally do not bind to pure PC membrane, according to all-atom simulations of various C2 domains [6] and experimental binding studies of C2_PTEN_ [29]. The difference in surface electrostatics also explains the observed differences in membrane-binding energy between C2 domains from PTEN and SHIP2 (Figure 4); C2 from PTEN has an overall positive charge on the surface facing the membrane, whereas C2 from SHIP2 also contains many negatively charged residues on the surface that faces the anionic membrane.

### 4.2. Comparison of Experimentally and Computationally Estimated Binding Energies

The free energy values we calculated are somewhat overestimated when compared with experimental estimates [29,30,31,32,33]. E.g., wild-type PTEN binding to a PC/PS/PIP_2_ (70:29:1) plasma membrane mimic was estimated to have a K_d_-value of 0.04 ± 0.01 µM, corresponding to a free energy of −42 ± 1 kJ/mol at 293 K [32]. By contrast, our calculations estimated a free energy value of −163 ± 1 kJ/mol for PTEN binding to a PC/PS/PIP_2_ (80:15:5) membrane. Part of this discrepancy may be ascribed to the higher content of PIP_2_ in the simulations compared with the experiment (5 mol% compared with 1 mol%), albeit PS content is lower in the simulations (15 mol% compared with 29 mol%). Moreover, it has recently been shown that the Martini 2 forcefield is prone to overestimate protein–lipid affinity for amphipathic helices [34], which may be the case for peripheral membrane proteins as well, albeit our previous simulations on C2 domains not indicating this [6]. When converting from coarse-grained to all-atomistic resolution, we saw an increase in the protein–membrane distance for both PTEN and SHIP2 (Figure 5B). Such loosening of the binding pose is in line with an overestimation of the binding energy in the coarse-grained MD simulations; albeit, we note that both proteins stayed bound at the membrane in the atomistic simulations.

Thus, while comparison of the absolute free energy values between our calculations and experiments is difficult, we are able to compare trends between them.

### 4.3. C2 Facilitates Membrane Binding in PTEN

It has been shown experimentally that the C2 domain of PTEN (C2_PTEN_) contributes to membrane binding on a PC/PS membrane with a K_d_ of 84 ± 9 µM (ΔG = −23 ± 1 kJ/mol) compared with a K_d_ of 2.9 ± 0.3 µM (ΔG = −31 ± 1 kJ/mol) for the full-length PTEN [29]. Our simulations show a large binding energy for C2_PTEN_, which is qualitatively consistent with this large contribution of C2_PTEN_ to the binding energy of PTEN, as determined experimentally. Moreover, in our simulations, Ptase_PTEN_ must have the C2_PTEN_ domain present to achieve its productive binding conformation, which confirms its role as a lipid anchor for PTEN.

In contrast to PTEN, the addition of the C2 domain to a Ptase construct of SHIP2 experimentally showed no increase in binding affinity to PIP_3_, but rather a modest decrease (K_d_ 30 ± 3 µM for Ptase_SHIP2_ vs. K_d_ 52 ± 4 µM for full-length SHIP2) [30]. This agrees qualitatively with the relatively low binding energy of C2_SHIP2_ in our simulations; albeit, we cannot explain the observed decrease in binding energy upon addition of C2_SHIP2_, which was observed experimentally [30]. Our simulations also show that the Ptase domain of SHIP2 can bind in its productive conformation without the C2_SHIP2_ domain, suggesting that the C2 domain does not act as a lipid anchor in SHIP2.

Thus, our results confirm that for PTEN, the C2 domain facilitates membrane binding and positioning the active site towards its substrate [8,29,35]. However, the results also raise the following question: What is the role of the C2 domain in SHIP2? Interestingly, phosphorylation of the C-terminus of PTEN (i.e., the “C2-end”) has previously been shown to allosterically affect PTEN phenotypes [36], suggesting that C2 acts both as a lipid anchor and as an allosteric regulator in PTEN. So, C2 may also act as an allosteric modulator in SHIP2.

### 4.4. C2 May Allosterically Affect Catalytic Activity in SHIP2

It has been shown by Le Coq et al. that C2_SHIP2_ is essential for cellular function [10], and the authors suggested that C2_SHIP2_ provides allosteric interdomain effects, which lead to an increase in the catalytic activity of the Ptase domain. With MD simulations, the authors demonstrated that an interaction between the C2 domain and R649 in the Ptase domain alter the dynamics of helices α5-7 (residues 615–643) and of loop L4 (residues 674–684). The authors proposed that the change in dynamics of L4 by C2_SHIP2_ is essential for catalytic activity by enabling L4 to alternate between an open and closed conformation [10]. Here, we performed a similar analysis, but with SHIP2 bound to a lipid membrane. Our simulations support that the presence of C2 in SHIP2 leads to conformational changes in the L4 loop and the α5-7 helices (Figure 6). However, we did not observe any change in flexibility in these regions. The presence of a lipid membrane quenched the change in dynamics of these regions and changed the dynamics of the L2 loop.

In summary, our simulations support the putative role of C2 in SHIP2 as an allosteric regulator of the Ptase domain.

## 5. Conclusions

Although SHIP2 and PTEN are both phosphatases and contain a catalytic Ptase domain and an adjacent C2 domain, the role of their C2 domains appears to be different. The C2 domain of PTEN participates significantly in membrane binding and is important for the positioning of the active site in the Ptase domain towards the lipid substrate—a role also observed for other C2 domains [37]. In contrast, the C2 domain in SHIP2 bound relatively weakly to anionic phospholipid membranes, and the Ptase domain was able to gain active conformation in the absence of the C2 domain. Therefore, the role of C2 in SHIP2 must be different from the membrane-anchoring role it undertakes in PTEN. Our simulations suggest that the C2 domain is responsible for allosteric interdomain changes in the L4 loop and in the α5-7 helices, as proposed previously [10]. This allosteric effect is likely to be essential for catalytic activity and can explain why the C2 domain is still essential for protein function in SHIP2.

## Figures and Tables

**Figure 2 membranes-13-00408-f002:**
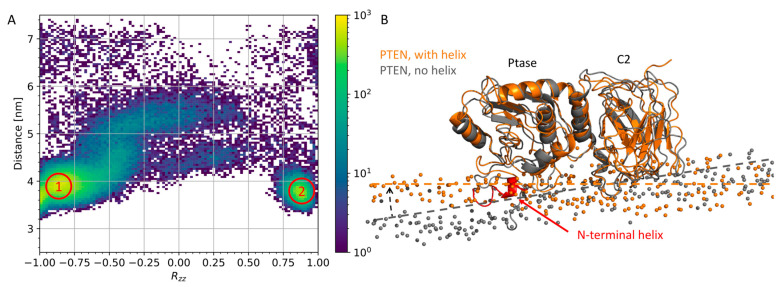
The role of the N-terminal helix of PTEN: (**A**) *R_zz_* vs protein–membrane center-of-mass distances for 25 repeats of coarse-grained simulations of full-length PTEN with N-terminal helix. Two binding modes were observed (red circles), with mode 1 being the productive mode, as shown in panel B, and mode 2 being unproductive, i.e., the active site of the Ptase domain was pointing away from the membrane (not shown). (**B**) Alignment of PTEN productive binding mode (orange, helix in red), with the productive mode of PTEN without the N-terminal helix (gray). Alignment was performed after conversion from coarse-grained to all-atom resolution. Bilayer planes are shown (dashed lines) together with phosphates from the lipid headgroups (spheres).

**Figure 3 membranes-13-00408-f003:**
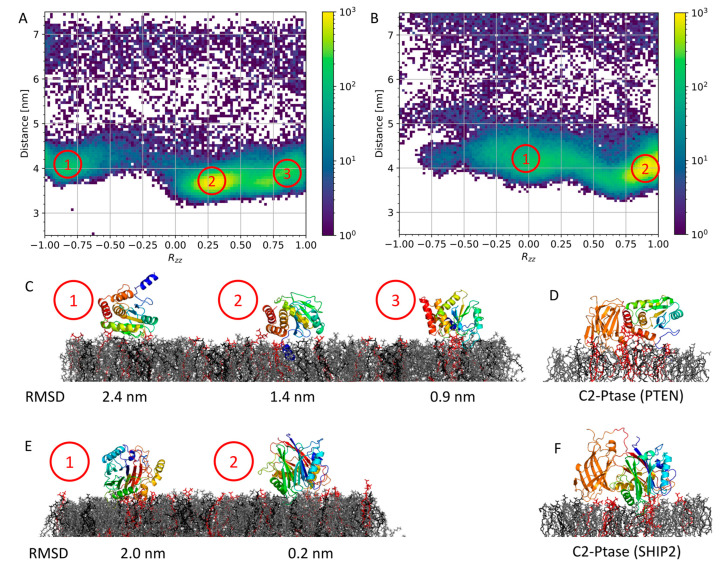
Simulations of isolated Ptase domains binding to lipid membrane: (**A**) Density map for 25 simulations of Ptase_PTEN_. The center-of-mass distance between Ptase_PTEN_ and the lipid bilayer is plotted against the orientation with respect to Ptase of full-length PTEN. *R_zz_* = 1 means the orientation is the same, and *R_zz_* = −1 corresponds to a 180° rotation. Dominant binding modes are numbered. (**B**) is as panel A but for simulations of Ptase_SHIP2_. (**C**) Dominant binding modes for Ptase_PTEN_, with corresponding RMSD values with respect to Ptase of the full-length PTEN simulation. (**D**) Dominant binding mode for full-length PTEN. (**E**) Dominant binding modes for Ptase_SHIP2_. (**F**) Dominant binding mode for full-length SHIP2. All proteins are shown in a rainbow color scheme, from blue in the N-terminus, over green, yellow, and orange to red in the C-terminus. PC lipids are gray, PS are black, and PIP_2_ are red.

**Figure 4 membranes-13-00408-f004:**
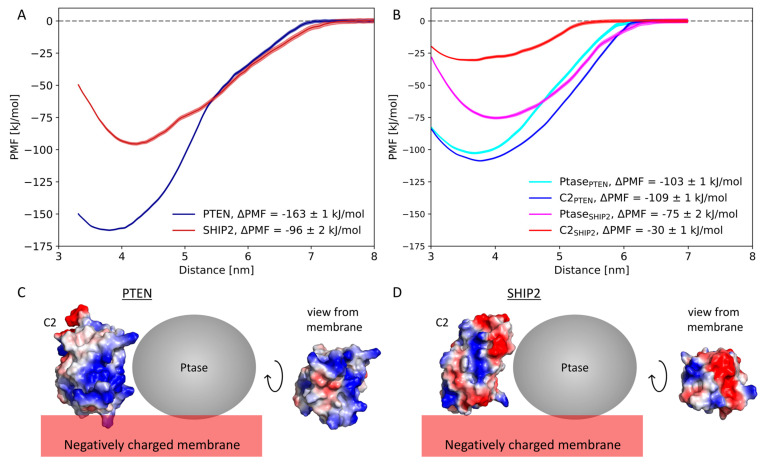
Binding energies as estimated by the potential of mean force: (**A**) Full-length PTEN (blue) and SHIP2 (red). Shaded areas show one standard deviation spread. (**B**) Ptase (light blue) and C2 (dark blue) from full-length PTEN. Ptase (purple) and C2 (red) from full-length SHIP2. Mean ± 1 standard error, estimated by bootstrapping [24]. Binding energy (ΔPMF) was defined as the difference between the minimum PMF value and the value in bulk. (**C**,**D**) Electrostatic surface of C2_PTEN_ (**C**) and C2_SHIP2_ (**D**), with blue indicating positive and red indicating negative charge. Positions of the negatively charged membrane and the Ptase domains are indicated schematically.

**Figure 5 membranes-13-00408-f005:**
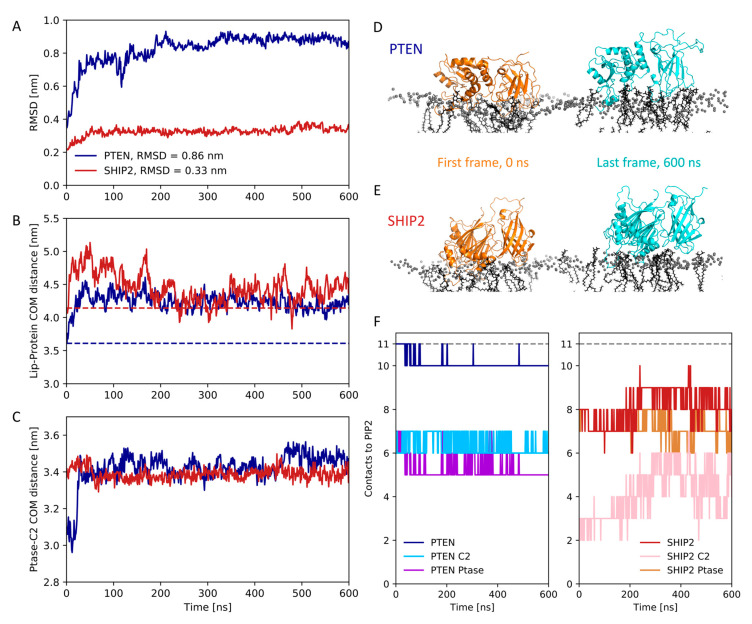
Atomistic simulations of full-length PTEN and SHIP2: (**A**) RMSD with respect to the binding poses from the coarse-grained simulations of PTEN and SHIP2. (**B**) Center-of-mass (COM) distance between membrane and proteins. Dashed lines show the distances for the initial coarse-grained binding poses. (**C**) Center-of-mass distance between C2 and Ptase domains for PTEN and SHIP2. (**D**) First (orange, 0 ns) and last (light blue, 600 ns) frames of the all-atom simulation of PTEN. Phosphate beads from the lipid heads are shown in gray. (**E**) First (orange, 0 ns) and last (light blue, 600 ns) frame from the SHIP2 all-atom simulation. (**F**) Number of contacts (with minimum distance less than 0.6 nm) between PIP_2_ lipids and PTEN, SHIP2 or their C2 or Ptase domains. There were 11 PIP_2_ lipids in the upper membrane leaflet, so this was the maximum possible number of contacts. A PIP_2_ can be simultaneously in contact with C2 and Ptase if positioned at the junction.

**Figure 6 membranes-13-00408-f006:**
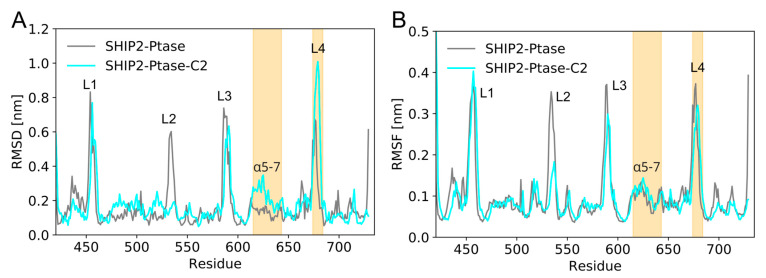
RMSD and RMSF of all-atom simulations of full-length SHIP2 and Ptase_SHIP2_: (**A**) RMSD of the Ptase section are calculated per residue as time average over the 600 ns simulations, with respect to the binding poses at 0 ns for Ptase_SHIP2_ (grey) and full-length SHIP2 (Ptase-C2, cyan). Loop L4 (residues 674–684) and helices α5-7 (residues 615–643) are highlighted in orange. (**B**) Corresponding RMSF.

## Data Availability

Simulated trajectories are available upon request. Scripts for reproducing the simulations are made available at GitHub: https://github.com/andreashlarsen/John2023-PTEN_SHIP2_C2, accessed on 23 March 2023.

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
