# Peer review of "The Role of C2 Domains in Two Different Phosphatases: PTEN and SHIP2"

_membranes, 2023, doi:10.3390/membranes13040408_

Round 1

Reviewer 1 Report

The manuscript titled “The role of C2 domains in two different phosphatases: PTEN and SHIP2” was prepared to compare the role of C2 domains in PTEN and SHIP2 proteins. The author found that compared to the C2 domain of SHIP2, the C2 domain of PTEN contributed to membrane recruitment significantly, and was necessity for the Ptase domain in gaining its productive membrane binding conformation. In contrast, C2 domain allosterically affects catalytic activity in SHIP2. These findings contribute to understanding the function of the C2 domain in phosphatases.The data of the manuscript basically supports the conclusion, but there are still some problems that need to be explained before the official publication.

1) In the methods section, the authors should provide the principles for setting the initial relative positions between protein and membrane. From the structural point of view(Figure 5D, 5F), the relative initial positions of PTEN and membrane seems to be somewhat different from that of SHIP2 and membrane,which is hard to judge whether it affects the analysis of C2 domain function.

2)From the results, the C2 domain showed different membrane-binding abilities in PTEN and SHIP2 proteins. Could the authors analyze the reason for this difference. For example, the difference between the two C2 domains as well as the difference between the relative position of the C2 domain and the Ptase domain in the two proteins.

3)Besides the RMSD and RMSF results, the authors also need to give more evidence to prove that the C2 domain regulates the catalytic activity of SHIP2.

4) It would be useful to provide some specific details of the combination between membrane and protein.

Author Response

We thank the reviewers for their valuable comments and suggestions. In the attached document, we address all comment. The corresponding changes are highlighted in the revised manuscript.

Reviewer 2 Report

This paper by John et al reports the coarse-grained (Martini) binding strengths of two phosphatases with and without their C2 domains, establishing model evidence and molecular detail for the relative impact on binding of the C2 domains: in particular, that the SHIP2 C2 domain contributes less to binding than that of PTEN. Active binding poses are found for both proteins with Martini and relaxed with all-atom forcefields. 

I believe the essence of the paper is valuable, that the C2 domain of PTEN appears to assist in the binding to anionic lipids (here PIP2) moreso than that of SHIP2. However, the interpretation of the magnitudes of the binding simply go too far, as I detail below. Instead, I hope the authors can focus on the sensitivity to the numbers of PIP2 bound (e.g., how effective are the two systems at collecting PIP2? There seem to be significant differences. The Martini predictions for the binding conformations of PTEN and SHIP2 should be valuable for the field, especially when considered in tandem with all-atom simulations showing some relaxation of the depth of insertion of PTEN. The corresponding author, AHL, is a promising young researcher who needs to take care for his reputation and talk seriously about the real deficiencies of the Martini forcefield for computing binding strengths.

Can the authors show the sequence differences (e.g., K/R residues) that lead to the varied binding of PIP2, or is it the pose?

Major criticism:

In a previous paper on C2 domains two of the authors (MSPS and AHL) write: “For that system, the energies were not overestimated compared with experimental values (but rather tended to be underestimated unless one allowed for multiple PIP molecules binding to each PH domain), which strengthens our justification for applying a comparable approach to the C2 system. We therefore suggest that the substantial binding energies observed are not an artifact from the force field.” 

Now for the comparison to the experiment they find that the experiment and simulation disagree on the Kd for PTEN binding to be off by 120 kJ/mol, that is, approximately 20 orders of magnitude in the amount of PTEN bound. They write: “it has recently been shown that the Martini 2 force field is prone to overestimate protein-lipid affinity“ and that only the relative trends should be taken seriously.

I am concerned that when results agree with experiment they argue that Martini does well, and when it does not agree perhaps only relative trends should be considered. 

I don’t think that it can be seriously argued that an experimental Kd going from 30 to 52 microM (a factor of less than two) is in qualitative agreement with a delta-G going from -30 to -109 kJ/mol (a factor of 10 trillion). I really think this is not forthright. At least compare the values in the same mode (Kd or delta-G) please.

Minor issue:

Somehow the number of PIP2 contacts between the C2 and Ptase domains did not show up in Figure 5F. This hobbled a bit my ability to interpret the PIP2 binding although it was clear from the sum-total PIP2 binding that PTEN gathered more PIP2.

Author Response

(The authors gave the same response as above.)
